# Assessment of Cracking Widths in a Concrete Wall Based on TIR Radiances of Cracking

**DOI:** 10.3390/s20174980

**Published:** 2020-09-02

**Authors:** Tung-Ching Su

**Affiliations:** Department of Civil Engineering and Engineering Management, National Quemoy University, 1 Da Xue Rd., Kinmen 892, Taiwan; spcyj@nqu.edu.tw; Tel.: +886-82-313-527

**Keywords:** thermal infrared radiances, width assessment of concrete crack, temperature gradients, image processing, statistical analysis

## Abstract

The techniques of concrete crack detection, as well as assessments based on thermography coupled with ultrasound, have been presented in many works; however, they have generally needed an additional source of thermal infrared (TIR) radiance and have only been applied in laboratories. Considering the accessibility of thermal infrared cameras, a TIR camera (NEC F30W) was employed to detect cracking in the concrete wall of an historic house with a western architectural style in Kinmen, Taiwan, based on the TIR radiances of cracking. An operation procedure involving a series of image processing and statistical analysis processes was designed to evaluate the performance of the TIR camera in the assessment of the cracking width. This procedure using multiple measurements was implemented from March to August 2019, and the t-tests indicated that the temperature differences between the inside and outline of the concrete cracks remained insignificant as the temperature or relative humidity (RH) in the subtropical climate rose. The experimental results of the operation procedure indicated that the maximum focusing range, which is related to the size of the sensor array, and the minimum detectable crack width of a TIR camera should be 1.0 m and 6.0 mm, respectively, in order to derive a linear regression model with a determination coefficient R^2^ of 0.733 to estimate the cracking widths, based on the temperature gradients. The validation results showed that there was an approximate R^2^ value of 0.8 and a total root mean square error of ±2.5 mm between the cracking width estimations and the observations.

## 1. Introduction

Concrete walls are commonly used as fences for private properties or building facades. However, cracks can appear in them due to thermal stress [1,2,3,4,5,6], inappropriate water-to-cement ratios [7] or inferior cement. Concrete walls can be separated into structural and non-structural walls. Cracking in non-structural walls does not affect structural safety but has an impact on the exterior and proofing performance. The chloride penetration behavior in concrete is largely affected by the width and number of cracks [8], and the resistance of cracked concrete against chloride ingress is mainly governed by the accumulated crack width and the water–cement ratio [9]. Thus, many works have presented inspection methods and technologies for the detection or assessment of cracks in concrete [10,11,12,13,14,15,16,17,18,19,20,21,22,23,24,25,26,27,28].

Most inspection methods/technologies for concrete cracks have been developed based on three kinds of images: visible color/gray-scale images, ultrasound and infrared thermography. Several image segmentation algorithms that are based on mathematic morphology or binarization have been proposed to detect cracks in concrete, and their detection accuracies have also been assessed [18,19,20,21]. With the recent rapid development of Artificial Intelligence (AI), machine learning methods have been widely applied to concrete crack detection based on visible images [22,23,24,25,26,27]. Kabir [27] applied the Artificial Neural Network (ANN) classifier to color, gray-scale and infrared thermal images to interpret three crack classes: a wide crack, a narrow crack and no crack. The experimental results indicated that infrared thermography produced more accurate results compared to the color and gray-scale images. Compared to the conventional ANN classification, deep learning neural networks, which usually consist of layers of alternative convolution and pooling and finally a full connection, have been successfully adapted to semantic segmentation for feature extraction and have been applied to concrete crack detection [22,23,24,25].

In addition to visible color/gray-scale images, ultrasound is considered to be an effective transmission medium to detect cracks in concrete. Quiviger et al. [12] investigated a real concrete macro-crack by using diffuse ultrasound and further simulated the propagation of diffuse ultrasonic energy in concrete in the presence of a real crack [13]. Zhao et al. [17] stated that there are limitations to the traditional ultrasonic tests for the generation and development of micro-cracks and used a literature review to identify nonlinear ultrasonic techniques as powerful methods for the initial damage detection of bonding materials and geo-materials. Ahn et al. [10] contended that diffuse ultrasound analysis shows good potential for field application in non-destructive testing and investigated the applicability of diffuse ultrasound for the evaluation of distributed micro-cracking damage in concrete.

Due to the anomalies that inhomogeneities impose on the temperature field, thermography specializes in subsurface damage identification [14]. Aggelis et al. [14] used infrared camera scanning to indicate the position of a crack and then placed ultrasonic sensors on specific parts of the surface to make a more detailed assessment of the depth of the crack. Jia et al. [11] used an ultrasound-excited thermography detection system to detect micro-cracks. The concept of the system is that ultrasonic energy is significantly attenuated and converted to heat when the ultrasound propagates through micro-cracks. Heat accumulates along the micro-cracks in the form of a temperature rise; therefore, the temperature field of the detected object can be observed with an infrared thermal imager. Furthermore, the characterization of small cracks with thermal imaging techniques has been presented, as have inverse models of crack estimation/description from the number of thermal images [15,16].

The formation of cracks is related to temperature and humidity changes [29]; therefore, crack inspections can be executed based on thermal infrared (TIR) radiances. Erenoglu et al. [30] constructed a 3D thermal model of a cultural heritage site in Turkey to show significant cracks resulting from water and wind erosion. Ibarra-Castanedo et al. [31] used an infrared camera to monitor the environmental thermal variations and evolution over the surface of an inspected bell tower due to solar irradiation and environmental temperature changes. An external crack on the inspected bell tower was identified in the principal component thermography-processed result. However, the above works, which are based on passive thermography, merely identify the appearances of cracks from the TIR models/images and fail to assess the crack sizes. On the contrary, several works have presented methods for crack detection and assessment based on active thermography coupled with ultrasound, but their methods need auxiliary heating sources rather than relying solely on self-emitted temperatures.

TIR images record the temperature values of an object by measuring the infrared radiation emitted by it. The recorded temperature differences can be used to show the qualitative differences in relative temperature so that the object can be distinguished from its surroundings [32]. Thus, this research considers that it would be feasible to assess concrete crack sizes from the recorded relative temperature differences (RTDs), which depend upon the emissivity difference between the concrete material and air in a concrete crack. Figure 1 shows a typical TIR image of a concrete crack, its corresponding 3D plot with a noise and the denoised 3D plot by pre-processing. As seen in Figure 1, the relative temperatures recorded within the crack pixels are mostly lower than those recorded within the other pixels. The above phenomenon is logical because the inside of a concrete crack usually has a lower solar irradiation and a higher environmental humidity. However, the atmospheric effects of the ambient temperature and relative humidity (RH) will influence the accuracies of receiving the radiations emitted not only from the object but also from other sources, such as the surrounding objects or the atmosphere [32,33]; thus, it is necessary to perform an experiment to quantify the effects [34]. In addition, if an RTD between the inside and outline of a concrete crack is significant, the concrete crack in a 3D plot will be displayed as a canyon-like shape. However, it is obvious that the 3D plot with dark noise cannot effectively display the canyon-like shape due to its lower signal-to-noise ratio [35] compared to the denoised plot.

A wider concrete crack should be indicated by a wider canyon-like shape, but it is certain that a temperature gradient (TG) must exist along with the RTD. The TG is a critical clue for the detection of subsurface defects in concrete [36]. To the best of our knowledge, there is no literature on the assessment of concrete cracks based on the TG changes resulting from the concrete crack. Moreover, the size assessment of sub-surface non-visible concrete cracks may be performed solely by a passive TIR camera, based on relative temperatures. In spite of the above concepts, there are three difficulties in applying a TIR camera based solely on relative temperatures to the size assessment of sub-surface non-visible concrete cracks: the first is the more challenging steps needed to calculate geometrical features in sub-surface concrete cracks directly from thermographs, as TIR images collect the radiation from the surface of concrete, which is the result of the heating of the volume; the second is the coarse spatial resolution of the TIR image, meaning that concrete crack morphologies cannot be effectively presented; and the last difficulty is that the true sizes of sub-surface non-visible concrete cracks are usually inaccessible, which hampers the assessment and estimation of the concrete crack. Thus, the TIR radiance-based concrete crack width estimation procedure proposed in this paper was preliminarily applied to the visible surface concrete cracks, the morphologies of which were identified by the corresponding RGB images of the TIR images.

Summarizing the statements above, in this paper, a two-phase-based procedure of concrete crack assessment was planned and the multiple measurements technique was implemented for the assessment of concrete cracks from March to August 2019. In the first phase, the influences of the atmospheric effects—i.e., the changes of ambient temperature and RH on the acquisition of the TIR images—were considered to determine the maximum focusing range and minimum detectable crack width. Based on the TIR images, which were acquired at the appropriate focusing ranges and able to show the detectable crack widths, in the second phase, the relationship between the crack widths and the TGs were identified for the crack width predictions.

## 2. Study Site and TIR Dataset

### 2.1. Study Site

In this study, the cracking concrete wall of an historic house with a western architectural style in Kinmen, Taiwan, was selected as the study site. The house was built in 1933 by a businessman, but it had been abandoned for many years. Due to the severe damage, the house was identified as a dangerous building, and therefore the Kinmen government began the implementation of a rehabilitation project. The cracking concrete wall belonged to one of the fences of the house, as shown in Figure 2.

According to an in-situ measurement, the widths of the crack in the concrete wall were between 1 and 25 mm. The ancient concrete wall merely supported its own weight; therefore, the probable cause of the cracks in the wall was the expansion and contraction of the cement due to temperature changes [37]. Kim et al. [2] considered factors such as TGs, drying shrinkage strain, and others to present an equation for the crack width prediction of thin fiber reinforced concrete overlays. Other studies have also discussed expansion/contraction models for concrete structures based on relative temperatures [38,39,40].

### 2.2. TIR Dataset

A TIR camera (NEC F30W) was employed in this study to acquire the images, including TIR and synchronous visible RGB images, for the cracking assessment. Due to the different sensors required for the TIR and visible RGB images, the spatial resolutions and the principal points of the two kinds of images were also different. Thus, the pre-processing of image registration was necessary before the TIR dataset could be analyzed. The relevant specifications, including the spectral range, spatial resolution, thermal sensitivity and temperature accuracy, of the TIR camera were 8 to 13 μm, 19,200 (160 × 120) pixels, 0.05 °C and ± 2 °C, respectively; however, the temperature accuracy was not assured at a focusing range of 0.5 m or closer. In this paper, the performance of the TIR camera during the cracking assessment at different focusing ranges of 0.5 m to more than 1.0 m was investigated.

Except for the month of July, the concrete crack underwent monthly monitoring from March to August 2019. Table 1 lists the related information regarding the acquisition of the TIR dataset. In order to improve the temperature accuracy, repeated imaging was implemented under certain fields of view (FOV). However, the expansion of the FOV would not only deteriorate the spatial resolution but also increase the number of mixed pixels [41], which are used record the multiple-scattering interacted TIR radiances of the concrete crack with its surroundings, thereby lowering the temperature accuracy. Therefore, different focusing ranges were tested on April 30 and June 20. For the other monitoring dates, minimum focusing ranges of 0.5 m—or slightly longer—were adopted in order to obtain the ideal spatial resolution and temperature accuracy. In addition, for certain FOVs, the different crack widths also resulted in different spatial resolutions and temperature accuracies. A smaller concrete crack indicated a more significant effect of a mixed pixel.

Temperature and RH are the main factors related to the process of weathering [42]. As the season changes from spring to summer, the temperature and RH gradually rise. The temperatures on March 12 and April 30 were higher than those on May 10; however, this may have been due to the TIR dataset being acquired in the afternoons. The interaction of temperature and RH during the different seasons promotes the physical weathering of masonry materials [42,43,44], thus influencing the inside as well as the outside temperatures of a concrete crack.

## 3. Methodology

This study involved two phases for the concrete crack assessment. In the first phase, the maximum focusing range and the minimum detectable crack width for the assessment were determined using a series of image processing and statistical analysis processes. In the second phase, the regression model for the detectable crack widths and the corresponding TGs was established. Figure 3 shows the proposed operation procedure of the concrete crack assessment.

In the first phase, the segmentation of the visible RGB images was executed to identify the regions of interest (ROIs) of the concrete crack in order to indirectly extract the temperatures within the ROIs from the TIR images. The image segmentation applied a weighted median filter consisting of 5 × 5 elements with a given weight value of 5 to the visible RGB images in order to smooth the environmental noise and maintain the textures of the ROIs [20,45,46]. Hereafter, the morphological operation technique of cross-curvature evaluation (CCE) [18] was applied to the filtered images in order to enhance the concrete crack for further segmentation according to the ROIs. By applying the Sobel operator [47] to the ROIs, the crack outlines could be successfully detected. Consequently, the temperature values on the crack edges could also be indirectly extracted by overlaying the edge detection results on the TIR images.

In the second phase, pre-processing based on image denoising was necessary to smooth the TIR dataset before the TG calculation. The TG calculation was similar to the slope calculation of the terrain. The 3D plot shown in Figure 1 indicates that a significant TG would lead to a deep concave. By overlaying the ROIs on the TG calculation results, the TGs within the concrete crack could be extracted for the establishment of the regression model between the TG and the crack width. Finally, the accuracy of the crack width estimation was validated via measurement in situ. The following sections describe the methods involved in the proposed operation procedure.

### 3.1. Image Registration

In order to superimpose the visible RGB images on the corresponding TIR images, pre-processing based on image registration was necessary. In this study, the commercial software TAS Version 24.6A [48] was employed to execute the image registration. Firstly, the visible RGB and TIR images were individually given two control points. The accuracy of the image registration was improved by choosing two control points that were as far apart as possible within the same image [49] and, as precisely as possible, between the different images corresponding to the same feature in situ. Due to the lenses of the visible RGB and TIR being mounted on the same camera, the axis systems of the lenses were orthogonal. Thus, the TAS software adopted two-dimensional conformal coordinate transformation (2-D CCT) for the image registration.

The 2-D CCT method requires that at least two control points must be known for the three basic steps of scale change, rotation and translation. Scale change refers to the scale of a visible RGB image being made equal to that of a TIR image by multiplying the axis system of the visible RGB image by the scale factor s. Rotation refers to an auxiliary axis system being constructed through the origin of the scaled axis system of the visible RGB image parallel to the axis system of the TIR image, in which the angle *θ* must rotate from the scaled axis system to the auxiliary one. Finally, the origin of the auxiliary axis system is transferred to the origin of the axis system of the TIR image. Based on the above description, the image registration is expressed as
(1)$$\left[\begin{array}{c} X_{\mathrm{TIR}}\\ Y_{\mathrm{TIR}} \end{array}\right]=s\left[\begin{array}{cc} \cos\theta & -\sin\theta \\ \sin\theta & \cos\theta \end{array}\right]\left[\begin{array}{c} X_{\mathrm{RGB}}\\ Y_{\mathrm{RGB}} \end{array}\right]+\left[\begin{array}{c} T_{X}\\ T_{Y} \end{array}\right]$$
where (*X*_TIR_, *Y*_TIR_) and (*X*_RGB_, *Y*_RGB_) are the axis systems of the TIR and visible RGB images, respectively, and (*T*_X_, *T*_Y_) are the translation factors.

### 3.2. Crack Enhancement by CCE

Before executing the weighted median filtering, the registered visible RGB images with 1.31M (1280 × 1024) pixels needed to be resized to equal the TIR images and translated into grayscale images. The CCE algorithm consists of a series of morphological operations, including closing, reconstruction, subtraction and addition. A filtered grayscale image is defined as having a range of [*I*_min_, *I*_max_] as a functional *F*: *R*^2^→[*I*_min_, *I*_max_] and considers a linear structural element as a functional *B*: *R*^2^→**B**, where **B** is a subset of the line segment. Except for the type of structural element, the related parameters of *B* also include the length and direction.

Closing is the first morphological operator in the CCE algorithm and is defined as
(2)$$\varphi_{B}\left(F\right)=\varepsilon_{B}\left(\delta_{B}\left(F\right)\right)$$
where δ*_B_* and ε*_B_* are the basic morphological operators; i.e., dilation and erosion, respectively. In other words, closing is an operation of dilation followed by erosion. Considering a certain length and direction of *B*, the pixel values must be maximized by the dilation of the linear structural element in order to gradually maximize the morphology of the concrete crack. Erosion is opposite to dilation and refers to gradually reducing the maximized morphology in order to avoid an over-dilation to reach conformity with the correct morphology of the concrete crack as far as possible.

The next step of our method was to produce the complements for both the closed images and the weighted median filtered images. In the complements, the higher pixel values became the lower values, and the lower values become the higher values. Based on the complements of the closed and filtered images, a morphological reconstruction was performed to identify the crack regions from the filtered images. Finally, an image subtraction of the complements of the closed images from the reconstructed ones was performed to enhance the crack regions. Considering *n* directions of the linear structural element, the crack enhancement results of multiple directions needed to be superimposed.

In this paper, a linear structural element with a length of 12 pixels and directions from 0° through 180°, using 10° intervals, was introduced into the morphological operations of the CCE algorithm. The steps of the Algorithm 1 were as follows:
**Algorithm 1**   **begininitialize**
*i* ← 0, linear structural element *B*_(12)*i*_, filtered grayscale image *F*   **for**
*i* ← *i* + 10   $\varphi_{i}\left(F\right)\leftarrow \;\left(F\;⊕{\;B}_{\left(12\right)i}\right)$ ⊙ $B_{\left(12\right)i}$ # image closing   $C_{\varphi i}\leftarrow$ ç($\varphi_{i}\left(F\right)$), $C_{F}\;\leftarrow$ ç(F) # image complement   **do**
*j* ← *j* + 1 # image reconstruction                 $h_{k+1}=\left(h_{k}⊕{\;B}_{\left(12\right)i}\right)\cap^{\;}{\;C}_{F}$   **until**
$h_{k+1}$ = $h_{k}$   *R_G_*($C_{\varphi i}$) = $h_{k+1}$   **end**   *R_G_*($C_{\varphi i}$) ← *R_G_*($C_{\varphi i}$) - $C_{\varphi i}$ # image subtraction (Top-hats transform)   $R_{G}\left(C_{\varphi }\right)$
$\leftarrow \;\sum_{i}R_{G}\left(C_{\varphi i}\right)$   **until**
*i* = 180   **end.**

After the crack segmentation by the CCE algorithm, Otsu’s technique, which is based on the discriminant analysis to help determine an optimal threshold for binary transformation [50], was employed to transform the CCE results from grayscale to binary and to derive the ROIs of the concrete crack.

### 3.3. Statistical Analysis

In this paper, an inter-quartile range (IQR) calculation was introduced to describe the extracted inside and outline temperature ranges of the concrete crack. Thus, the influences of the focusing ranges and crack widths on the ranges between the 75th and 25th percentiles—i.e., the IQRs—as well as the median differences between the inside and outline temperatures of the concrete crack were discussed.

In addition, in order to evaluate the significance of RTD between the inside and the outline of the concrete crack, one-tailed and two-tailed t-tests were performed. In the two-tailed t-test, the null (*H*_0_) and alternative (*H*_1_) hypotheses were expressed as
(3)$$\{\begin{array}{c} H_{0}:T_{i}=T_{o}\\ H_{1}:T_{i}\neq T_{o} \end{array}$$
where *T_i_* and *T_o_* are the inside and outline temperatures of the concrete crack, respectively. If the two-tailed t-test showed that *T_i_* and *T_o_* were significantly different—i.e., *H*_1_ was accepted—the one-tailed t-test would then be performed and expressed as
(4)$$\{\begin{array}{c} H_{0}:T_{i}=T_{o}\\ H_{1}:T_{i}>T_{o} \end{array}\;\mathrm{or}\;\{\begin{array}{c} H_{0}:T_{i}=T_{o}\\ H_{1}:T_{i}<T_{o} \end{array}$$

Referring to the illustration of Figure 1, an acceptance of $T_{i}<T_{o}$ instead of $T_{i}>T_{o}$ was expected. Finally, the maximum focusing range and the minimum detectable crack width was determined by concluding the IQR calculation and the t-test results.

### 3.4. Denoising for TIR Image

Budzan and Wyżgolik [51] indicated that the noise sources of TIR images are mainly from TIR sensors and the interference of the signal processing circuit. The non-uniformity of focal plane array (FPA) uncooled microbolometers results from different response (gain and offset) of the FPA pixels due to the fabrication process. The noise in TIR images is not only determined by the detector but also by background and emissivity fluctuations of the object. For example, in the TIR image shown in Figure 1, the recorded temperature values in one array of the column direction are shown as the blue polyline in Figure 4. A severe fluctuation, which hampers the calculation of the TG, can be seen; thus, a denoising process was necessary for the observed temperature values. Among the signal oscillations, the largest oscillation results from the RTD between the inside and the outline of the concrete crack.

Different kinds of filters, based on the spatial or frequency domains of images, have been proposed in many works for the denoising of visible, infrared, computed tomography and magnetic resonance imaging [52,53,54,55]. Frequency domain-based filtering tools, also known as transform domain filtering tools [53], are applicable as digital signal processing tools in both the 1-D and 2-D domains. The discrete wavelet transform (DWT) is a widespread transform domain filtering tool in which most of the information in the image is encoded in a small number of high-valued coefficients [56,57]. DWT depends on a threshold in order to differentiate between signal and noise. Discrete wavelet threshold denoising can be used to perform a multi-scale and multi-resolution analysis on noisy images with small root mean square errors and high signal-to-noise ratios; however, signal oscillations during hard threshold denoising, as well as constant deviations during soft threshold denoising, are inevitable in traditional wavelet threshold denoising methods [55]. In [53], the current image denoising methods, based on transform domain filtering, were reviewed and compared. DWT—a transform domain filtering method—showed superior performance to the other methods, such as the Fourier transform and the discrete cosine transform. Moreover, multi-resolution-based algorithms perform far better than single-resolution algorithms.

For the denoising process, in this paper, a three-level 2-D Daubechies wavelet decomposition approach—the most popular set of DWTs [58]—was introduced to construct an approximation level of a TIR image at coarser levels using low-pass filter banks and at a detailed level using high-pass filter banks. Each level of approximation was further decomposed into an image representation at the next scale. Consequently, a vector containing the detail coefficients could be obtained from the constructed wavelet decomposition structure, which was composed of another vector with the same length as that containing the detail coefficients and which contained the computed wavelet transform coefficients and a bookkeeping matrix that defined the arrangement of the above wavelet transform coefficients by soft thresholding [59]. Among the three levels, the corresponding thresholds to the first, second and third levels were given as 30, 25 and 20, respectively. Finally, the denoised TIR image was obtained by a wavelet reconstruction based on the vector containing the detail coefficients and the bookkeeping matrix. The red polyline in Figure 4 indicates the denoised temperature values. It can be seen that the severe fluctuation was greatly mitigated.

### 3.5. TG Calculation

In this paper, the TGs were calculated by applying a Sobel kernel to each pixel of the denoised TIR image for discrete difference detection between the rows and columns of a 3 × 3 pixel segment of the image. The larger the discrete difference, the higher the TG, which is defined as the vector $\nabla T=\left[\begin{array}{c} g_{x}\\ g_{y} \end{array}\right]=\left[\begin{array}{c} \frac{\partial T}{\partial x}\\ \frac{\partial T}{\partial y} \end{array}\right].$ The magnitude of ∇T at the center pixel *z*_5_ is ${\left[g_{x}^{2}+g_{y}^{2}\right]}^{0.5}={\left\{{\left[\left(z_{7}+2z_{8}+z_{9}\right)-\left(z_{1}+2z_{2}+z_{3}\right)\right]}^{2}+{\left[\left(z_{3}+2z_{6}+z_{9}\right)-\left(z_{1}+2z_{4}+z_{7}\right)\right]}^{2}\right\}}^{0.5},$ where $z_{1}$ through $z_{9}$ sequentially indicate the denoised temperature values at the pixels, from left to right and top to bottom, of the 3 × 3 kernel.

## 4. Results and Discussion

### 4.1. Descriptive Statistics for the Inside and Outline Temperatures of the Concrete Crack

The IQR calculation results for the multiple measurements are shown in Figure 5. In this research, certain scenes of interest were repeatedly imaged to investigate the influence of random errors. The obtained IQRs of the repeated imaging revealed fluctuations such as those found in image IDs 1–3 and 5–8 on March 12. At a fixed focusing range of 0.5 m, the IQRs seemed to be positively related to the crack widths. As shown in Figure 5a, based on the inside temperatures of the concrete crack, the IQRs of image IDs 1–3 were slightly larger than those of image IDs 5–8, because image IDs 1–3 showed wider crack sections. On 10 May, a positive relationship was found between the crack widths and the IQRs of the inside temperatures of the concrete crack. However, on August 14, there was no significant difference among the IQRs of the inside temperatures of the concrete crack. This was most likely caused by the rising temperature and the RH in summer (see Figure 5i). According to a technical report from FLIR Systems, Inc., atmospheric effects—one of the five factors influencing radiometric temperature measurements—absorb and emit infrared radiation based on the air density, air temperature, RH and distance between the object’s surface and the camera [32]. The atmospheric effects could be largely negated by making measurements to within 10 m or less of the target surface and in a cool and clear atmospheric setting. Table 2 presents a numerical description of the boxplots shown in Figure 5, in which the shadowed image IDs indicate the repeated imaging of certain scenes.

It can be seen in Figure 5a,b that image IDs 9 and 10 had the largest and smallest IQRs, respectively. Image ID 9 recorded crack sections of between 16 and 17 mm in width, which were wider than the crack sections recorded in the other images. Image ID 10 recorded the entire crack; therefore, a focusing range of much larger than 0.5 m was needed. Nevertheless, the large focusing range allowed image ID 10 to display finer cracks than the other images, thus presenting the smallest IQR.

In addition to the focusing range of 0.5 m, focusing ranges less than and greater than 1.0 m were tested on April 30 and June 20, respectively. The wider crack sections seemed to be associated with larger IQRs for the inside temperatures of the concrete crack; however, the longer focusing ranges led to smaller IQRs. Furthermore, the higher temperatures and RH in the summer may have reduced the IQRs. On April 30, image IDs 8–10 and 5–7 were acquired at focusing ranges of 0.6 and 0.7 m, respectively. These images showed crack widths ranging from 2 to 7.5 mm and from 3 to over 20 mm, respectively. Despite the slightly longer focusing range of image IDs 5–7 compared to image IDs 8–10, image IDs 5–7 recorded much larger crack widths; therefore, they obtained larger IQRs than image IDs 8–10 (see Figure 5c).

A comparison of the IQRs between the inside and outline temperatures of the concrete crack in the spring (i.e., March through May 2019) showed that most of the IQRs of the inside temperatures were larger than those of the outline temperatures. Nevertheless, in summer, the IQRs between the inside and outline temperatures of the concrete crack were approximately the same. This result illustrated that, at a lower temperature or RH, the cracking caused an obvious temperature distribution difference on the concrete wall. As the temperature or RH rose, the amount of the difference decreased. In addition, the obtained median differences between the inside and outline temperatures of the concrete crack could also be demonstrated to support the above statement. As expected, most of the median differences were positive, which indicated that the inside temperatures were nearly always lower than the outline temperatures. However, in June and August 2019, due to the high temperatures and RHs, the cracking was insufficient to cause a significant RTD on the concrete wall.

### 4.2. Significance Test of RTD between the Inside and Outline of the Concrete Crack

As the data size of the inside temperatures was larger than that of the outline temperatures, a batch-by-batch t-test was necessary to reveal the significance of the RTD between the inside and outline of the concrete crack. Assuming that there were *n* inside temperature values (*T_i_*) and *m* (*n* > *m*) outline temperature values (*T_o_*) in a temperature distribution matrix, *l* (*l* = *m*) *T_i_* values as a batch were withdrawn in sequence from the top left to bottom right of the matrix and arranged as a vector. At the same time, the *m T_o_* values were also arranged as a vector, according to the above approach. In other words, the *n T_i_* values were divided into several batches, usually with a surplus. The vector length of the *T_i_* values of each batch needed to be the same as that of the *T_o_* values in order to facilitate the implementation of the t-test. As for the surplus *T_i_* values, they were also used to perform the t-test with the *T_o_* values, which had the same number as the surplus *T_i_* values, but were reversely withdrawn from the bottom right to top left of the temperature distribution matrix.

Table 3 shows the t-test results for the RTDs, given a significance level of α = 0.05. Most of the obtained *p* values were approximately 0, thereby allowing us to accept hypothesis *H*_1_ (*T_i_* > *T_o_* or *T_i_* < *T_o_*). Accepting that *T_i_* > *T_o_*, as presented in hypothesis *H*_1_, would mean that the *T_i_* values within some batches were significantly higher than the entire *T_o_* values; however, this result was not expected.

As shown in Table 3, there was also a great proportion of *T_i_* values corresponding to hypothesis *H*_0_ (*T_i_* = *T_o_*). In this paper, it was found that there were three critical factors related to the acceptance of hypothesis *H*_0_: firstly, the rising seasonal temperatures and RH values would result in an insignificant RTD; secondly, the areas of the concrete crack with small widths usually had approximate *T_i_* and *T_o_* values—in other words, the thermal sensitivity of the TIR camera was insufficient for the temperature sensing of small concrete cracks—and finally, the long focusing range caused the concrete crack to be zoomed out in the image and to be out of the thermal sensitivity range of the TIR camera.

For a dataset with repeated imaging, the t-test results among the images should be similar so that the stability of the TIR camera for crack assessment can be demonstrated. In spite of this, there were a number of datasets with repeated imaging that had inconsistent t-test results. For example, on March 12, the t-test results of image IDs 5 and 6 were not only greatly different from each other but also from those of image IDs 7 and 8, and on May 10, the t-test result of image ID 12 was also greatly different from those of image IDs 11 and 13. The inconsistent t-test results may have been the result of random errors; however, extreme testing results could be effectively identified by repeated imaging.

### 4.3. Determination of Maximum Focusing Range and Minimum Detectable Crack Width

As seen in Table 2, on June 20, the values or averaged values of Med*_To_*–Med*_Ti_* were mostly approximate to 0 °C in spite of the greatly different crack widths. Moreover, as seen in Table 3, the t-test results on June 20 showed that most of the *T_i_* values corresponded to hypothesis *H*_0_ (*T_i_* = *T_o_*). Except for image ID 1, the other images on June 20 were acquired at a focusing range of above 1.0 m. On May 10 and August 14, the values or averaged values of Med*_To_*–Med*_Ti_* were also mostly approximate to 0 °C; however, some of the *T_i_* values accepted corresponded to *H*_1_ (*T_i_* < *T_o_*), which may have resulted from the shorter focusing ranges than those on June 20. Consequently, it was found that, when considering the insignificant RTDs, due to the rising seasonal temperature or RH values, the maximum focusing range should not be above 1.0 m.

As seen in Table 3, the *T_i_* values in image ID 9 on March 12 and image ID 1 on April 30, which were used to investigate the larger crack widths above 11 mm, could be regarded as robustly supporting hypothesis *H*_1_ (*T_i_* < *T_o_*). Unfortunately, along with the rising seasonal temperature and RH values, the robustness of support of hypothesis *H*_1_ (*T_i_* < *T_o_*) decreased, as seen in the *T_i_* values for image ID 1 on June 20 and image IDs 1–3 on August 14; those images also recorded larger crack widths above 11 mm (see Table 2). On May 10, as seen in image IDs 6–10 in Table 3, crack sections with a maximum width of 6.5 mm had greater than 75% of the *T_i_* values, thus supporting hypothesis *H*_1_ (*T_i_* < *T_o_*). However, the *T_i_* values for crack widths less than 4.0 mm, such as those seen in image IDs 1–5, robustly supported hypothesis *H*_0_ (*T_i_* = *T_o_*). Thus, in this paper, it was conservatively assumed that the minimum detectable crack width should be 6 mm.

A review of the data in Table 1, Table 2 and Table 3 revealed that higher temperature and RH values, such as those found in June and August 2019, hampered the crack assessment using the TIR camera. On 20 June and 14 August, the concrete wall was facing the morning sun while the images were being acquired. The severe reflections from the concrete wall influenced the reflections from the concrete crack itself; therefore, the solar azimuth angle should also be considered when assessing concrete cracks based on TIR.

### 4.4. Regression Model Establishment between TG and Concrete Crack Width

According to the discussion in Section 4.3, the regression model considered the TG values of the crack sections with a width above 6 mm on the TIR images at focusing ranges within 1.0 m. A total of 105 crack sections were selected from all TIR images on 12 March and from the first seven TIR images on 30 April for the establishment and validation of the regression model, of which 49 crack sections were used for establishment and the other 56 were used for validation.

Taking image ID 7 on 30 April as an example (see Figure 6), the larger crack section induced a higher TG value. A seen in Figure 6, the peak values of the large crack section on the TG map usually existed near to the detected crack outlines. Due to both sides of the crack section being detected as outline pixels, this paper introduced two 3 × 3 kernels with the central positions located at the TG peak values on both sides to calculate a representative TG value for the crack section. Consequently, an average of the 18 TG values within the two kernels was obtained to represent the TG resulting from the crack section.

The established regression model and the validations for the estimated crack widths are shown in Figure 7, revealing a linear and positive correlation between the TG value and the crack width. The regression model, with a determination coefficient (R^2^) of 0.733, was expressed as **Y** = 0.0915**X** + 0.0457, where **X** and **Y** indicate the crack width and TG value, respectively. Figure 7b,c show the validation results for the crack sections based on the TIR images from 12 March and 30 April, respectively. Both of the validation results for 12 March and 30 April could obtain approximate R^2^ values above 0.8 and a total root mean square error (RMSE) of ±2.5 mm. In this paper, the crack widths of the validated crack sections were further separated into three categories—namely 6~10 mm, 11~15 mm and above 15 mm—and then calculated the RMSEs for each category (see Table 4). An RMSE (also regarded as the proportional error) of approximately ±2.0 mm was obtained for the different crack width ranges. Theoretically, the proportional error varies with the observed value, and the larger the observed value is, the larger the proportional error will be. However, as shown in Table 4, the obtained RMSEs did not increase along with the observed crack width ranges. In other words, in this paper, it was demonstrated that superior performance was obtained for width estimation for large crack sections compared to that for small crack sections.

## 5. Conclusions

In this paper, an operation procedure for concrete crack assessment based on thermal infrared radiances was proposed to evaluate the performance of a TIR camera (NEC F30W) in terms of sensing the cracking in a concrete wall. Focusing ranges from 0.5 to 1.9 m were tested using the TIR camera, and the concrete crack widths from 1.0 to 25.0 mm were investigated using multiple measurements from March to August 2019. From the theoretical research and experimental analyses, the following conclusions can be drawn:(1)The descriptive statistics of the inter-quartile range, coupled with the t-test, indicated that the difference between the inside and outline temperatures of the concrete crack would be insignificant as the seasonal temperature or relative humidity rose. In this study, multiple measurements were recorded from March to August 2019 in the subtropical climate of Kinmen, Taiwan. In future, multiple measurements should be recorded continuously over a number of years to observe the influence of climate change on the difference between the inside and outline temperatures of concrete cracks.(2)Wavelet-based image denoising is useful to smooth the TIR signals and facilitate the calculation of temperature gradients. The filtered waveform of the TIR radiances of a crack section (see Figure 4) was similar to the shape of the crack section. Based on the differences between the maximum and minimum temperatures of the crack sections, future regression models should also be established to estimate the depths of the crack sections.(3)The proposed operation procedure, based on a series of image processing and statistical analysis processes, not only effectively determined the maximum focusing range and minimum detectable crack width (1.0 m and 6.0 mm, respectively) but also successfully established a linear regression model between the temperature gradient and the crack width, with an R^2^ value of 0.733. Once a high-level TIR camera with a sufficient spatial resolution and thermal sensitivity for identifying concrete crack morphologies is obtained, the width estimation of sub-surface non-visible concrete cracks will be tested if the true widths are known.(4)According to the established linear regression model, the widths of several crack sections in the TIR images that were acquired in March and April 2019 were estimated. The derived R^2^ values between the estimated and observed crack widths were above 0.8, and the total RMSE was approximately ±2.5 mm. Nevertheless, in this paper, it was demonstrated that there was a greater proportional error for the width estimations of small crack sections than for large crack sections. A TIR dataset with a high spatial resolution is needed to reduce the proportional error of the width estimations of small crack sections, especially for detecting crack widths that are less than the current minimum limitation of 6.0 mm.

## Figures and Tables

**Figure 1 sensors-20-04980-f001:**
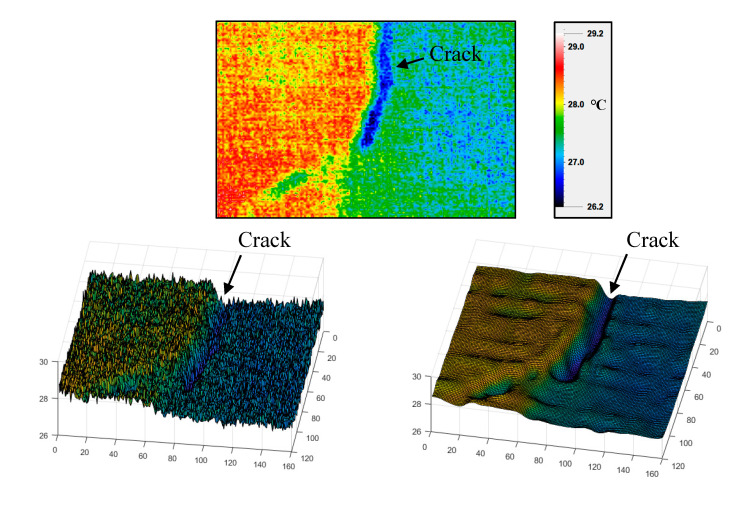
Thermal infrared (TIR) image of a concrete crack (**top**), thecorresponding 3D plot with noise (**bottom left**) and denoised 3D plot (**bottom right**).

**Figure 2 sensors-20-04980-f002:**
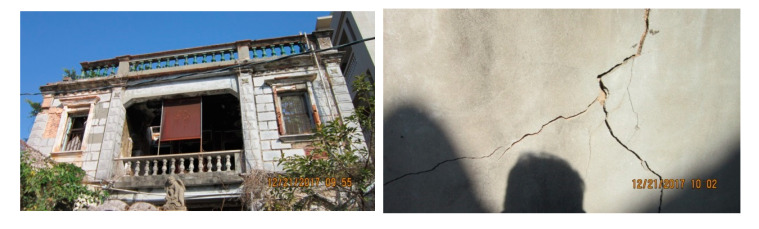
Historic western architectural house (**left**) and its damaged concrete wall (**right**).

**Figure 3 sensors-20-04980-f003:**
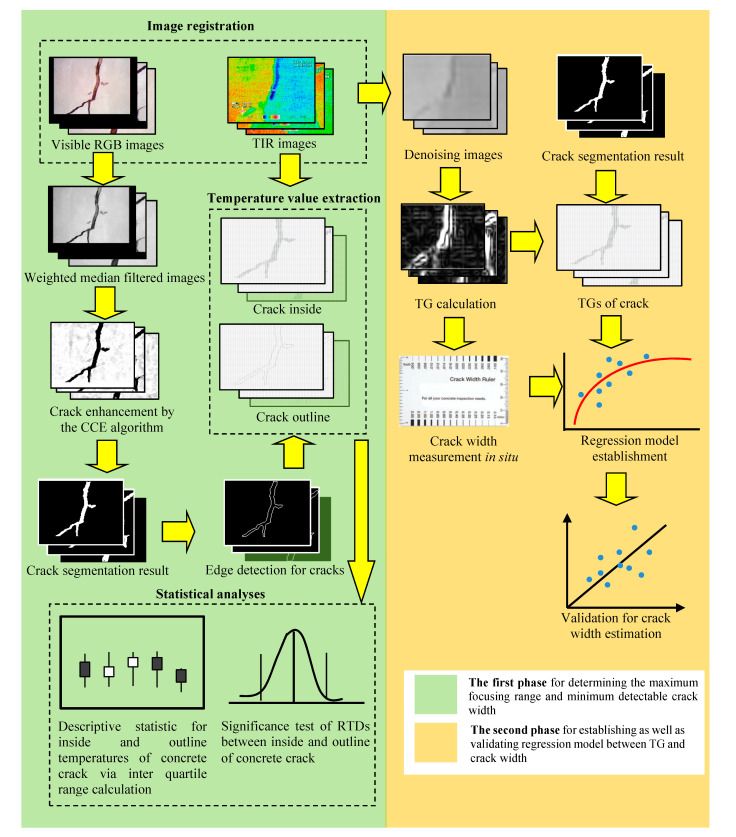
Proposed concrete crack assessment operation procedure. RTD: relative temperature difference; TG: thermal gradient.

**Figure 4 sensors-20-04980-f004:**
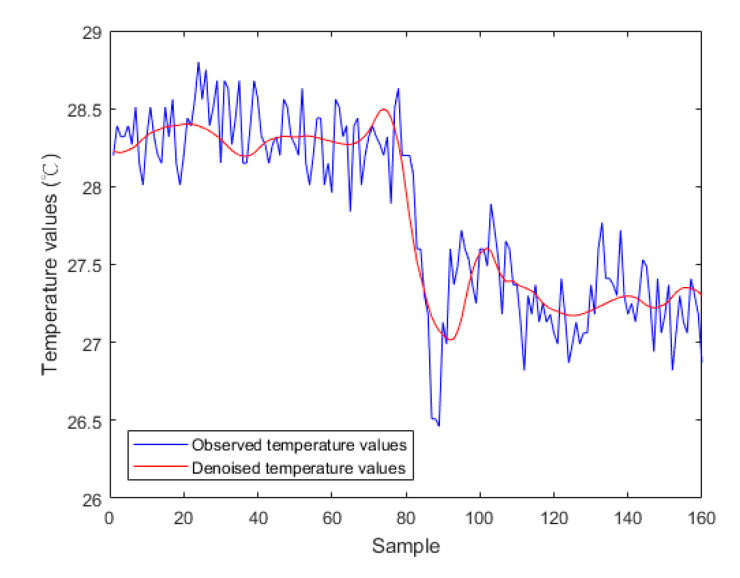
Temperature values in the TIR image array before and after image denoising.

**Figure 5 sensors-20-04980-f005:**
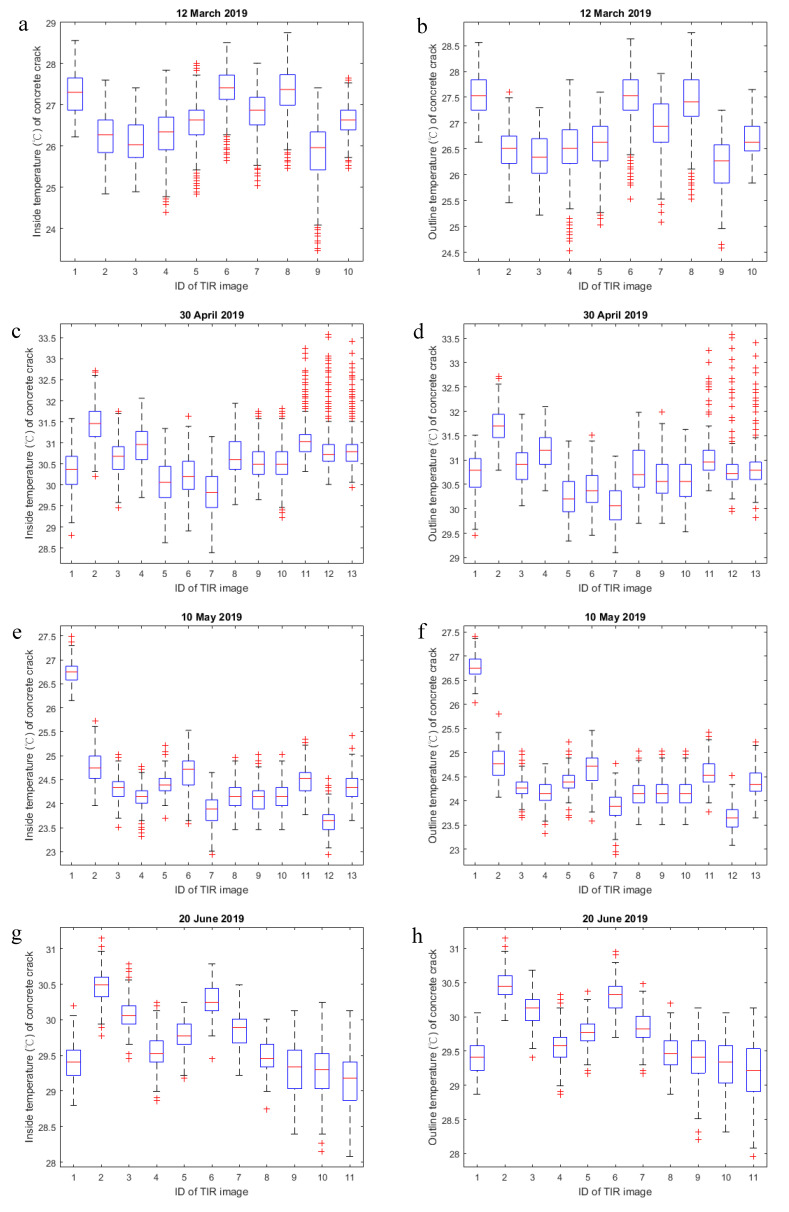
Inside (**left column**) and outline (**right column**) temperature ranges of the concrete crack during multiple measurements.

**Figure 6 sensors-20-04980-f006:**
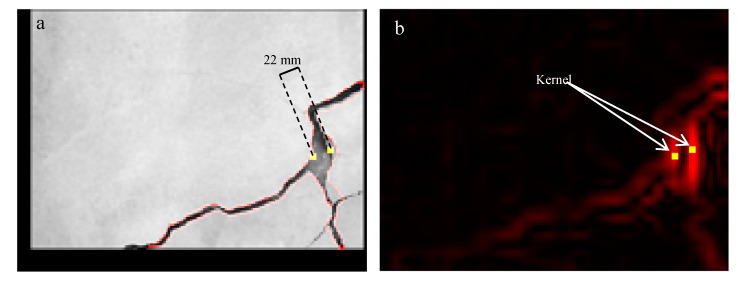
TG peak values of the concrete crack used for the establishment of the regression model: (**a**) a large crack section with a width of 22 mm, used as an example to illustrate the TG peak values; (**b**) the TG peak values of the large crack section existing within two 3 × 3 kernels, which were usually near to detected crack outlines, as shown by the red polylines in (**a**).

**Figure 7 sensors-20-04980-f007:**
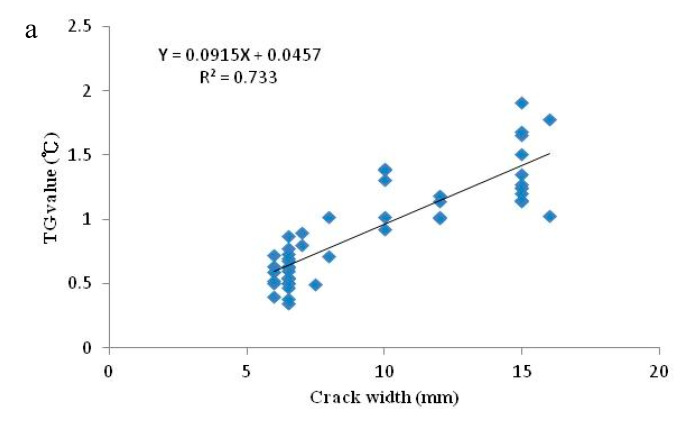
Estimation and validation of concrete crack widths: (**a**) the derived regression model between the TG value and the crack width; (**b**) and (**c**) the validation of the estimated crack widths on TIR images from March 12 and April 30 2019, respectively.

**Table 1 sensors-20-04980-t001:** Related TIR dataset acquisition information.

Date	Time	No. of TIR Images	Focusing Ranges (m)	Weather	
				Temperature (°C)	Relative Humidity (%)
12 March 2019	14:40~15:00	10	0.5 *	24.1	49
30 April 2019	15:30~15:50	13	0.5, 0.6, 0.7,0.8	27.8	78
10 May 2019	10:50~11:10	13	0.5	22.7	71
20 June 2019	08:20~08:40	11	0.5, 1.1, 1.3, 1.5, 1.9	28.1	85
14 August 2019	08:40~09:00	12	0.6	28.9	86
Total		59			

* Except image ID 10.

**Table 2 sensors-20-04980-t002:** Numerical description of the boxplots in Figure 5.

Date	Image ID	Focusing Range (m)	RCW * (mm)	*T_i_*		*T_o_*		Med*_To_*—Med*_Ti_*(°C)	Avg. of Med*_To_*–Med*_Ti_* (°C)
				IQR (°C)	Avg. IQR (°C)	IQR (°C)	Avg. IQR (°C)		
12 March 2019	1	0.5	6~15	0.78	0.79	0.59	0.60	0.23	0.26
	2			0.79		0.53		0.24	
	3			0.79		0.67		0.31	
	4		5~15↑	0.79	-	0.65	-	0.17	-
	5		2~11	0.60	0.65	0.67	0.68	0.00	0.06
	6			0.59		0.59		0.12	
	7			0.67		0.74		0.07	
	8			0.74		0.71		0.04	
	9		16~17	0.92	-	0.74	-	0.31	-
	10	0.5 ↑	2~20↑	0.48	-	0.48	-	0.00	-
30 April 2019	1	0.8	11~23	0.67	-	0.59	-	0.42	-
	2	0.5	6~15	0.60	0.60	0.48	0.53	0.24	0.24
	3			0.54		0.55		0.23	
	4			0.67		0.55		0.24	
	5	0.7	3~20↑	0.74	0.72	0.62	0.59	0.14	0.18
	6			0.67		0.55		0.17	
	7			0.74		0.60		0.24	
	8	0.6	2~7.5	0.66	0.58	0.76	0.67	0.10	0.08
	9			0.54		0.59		0.07	
	10			0.54		0.66		0.07	
	11	0.5	4~5	0.41	0.40	0.41	0.36	−0.07	−0.02
	12			0.40		0.31		0.00	
	13			0.40		0.36		0.00	
10 May 2019	1	0.5	1~3.5	0.29	0.36	0.31	0.35	0.00	−0.02
	2			0.47		0.50		0.03	
	3			0.31		0.24		−0.07	
	4		1~4	0.26	0.26	0.33	0.30	0.00	0.00
	5			0.26		0.26		0.00	
	6		1.5~6.5	0.50	0.47	0.47	0.42	0.00	0.00
	7			0.43		0.38		0.00	
	8		1.5~4	0.38	0.38	0.36	0.37	0.00	0.00
	9			0.38		0.38		0.00	
	10			0.38		0.38		0.00	
	11		2.5~3.5	0.38	0.36	0.38	0.37	0.00	0.00
	12			0.31		0.36		0.00	
	13			0.38		0.38		0.00	
20 June 2019	1	0.5	15~19	0.36	-	0.36	-	0.00	-
	2	1.9	6~19	0.28	-	0.28	-	−0.05	-
	3	1.5	6~20	0.26	-	0.31	-	0.07	-
	4	1.5	4~20↑	0.29	-	0.29	-	0.05	-
	5	1.3	1.5~6.5	0.29	-	0.24	-	0.00	-
	6	1.1	1~3.5	0.31	0.32	0.31	0.32	0.07	0.00
	7			0.34		0.31		−0.07	
	8			0.31		0.35		0.00	
	9	1.1	4~15	0.55	0.53	0.47	0.55	0.07	0.05
	10			0.50		0.55		0.04	
	11			0.54		0.62		0.04	
14 August 2019	1	0.6	11~25	0.35	0.32	0.38	0.33	0.00	0.00
	2			0.31		0.38		0.00	
	3			0.29		0.24		0.00	
	4		3~12	0.43	0.38	0.36	0.34	0.00	0.04
	5			0.36		0.36		0.00	
	6			0.36		0.31		0.12	
	7		1.5~6.5	0.36	0.32	0.40	0.33	0.00	0.00
	8			0.31		0.31		0.00	
	9			0.28		0.28		0.00	
	10		2.5~3.5	0.29	0.32	0.29	0.29	0.00	0.01
	11			0.35		0.28		0.00	
	12			0.31		0.29		0.03	

Note: RCW is the recorded crack width; * refers to measurements made in situ using a crack width ruler; *T_i_* is the inside temperature of the crack; *T_o_* is the outline temperature of the crack; IQR is the inter-quartile range; Med*_Ti_*, is the median of the inside temperature of the crack; Med*_To_* is the median of the outline temperature of the crack; The shadowed image IDs indicate the repeated imaging of certain scenes.

**Table 3 sensors-20-04980-t003:** T-test results for RTDs between the inside and outline of the concrete crack.

Date	Image ID	Pixel Number of Extracted Crack Temperature Values		Proportions of *T_i_* Values Accepting Null (*H*_0_) and Alternative (*H*_1_) Hypotheses (%)		
		*T_i_*	*T_o_*	*H*_0_: *T_i_* = *T_o_*	*H*_1_: *T_i_* > *T_o_*	*H*_1_: *T_i_* < *T_o_*
12 March 2019	1	1277	361	43.4	0.0	56.6
	2	1083	322	29.8	0.0	70.2
	3	964	314	2.3	32.6	65.1
	4	1211	446	0.0	26.3	73.7
	5	1423	481	66.2	33.8	0.0
	6	1466	837	0.0	0.0	100.0
	7	932	428	54.1	0.0	45.9
	8	1189	638	46.3	0.0	53.7
	9	1324	251	5.2	19.0	75.8
	10	728	376	0.0	51.6	48.4
30 April 2019	1	699	224	3.9	0.0	96.1
	2	1461	377	22.6	0.0	77.4
	3	1483	377	0.0	0.0	100.0
	4	1534	378	1.4	0.0	98.6
	5	796	331	16.8	0.0	83.2
	6	753	336	10.8	0.0	89.2
	7	815	347	14.8	0.0	85.2
	8	820	370	0.0	45.1	54.9
	9	826	392	5.0	47.5	47.5
	10	810	398	0.0	49.1	50.9
	11	644	252	100.0	0.0	0.0
	12	638	243	76.2	0.0	23.8
	13	604	226	37.4	37.4	25.2
10 May 2019	1	335	335	100.0	0.0	0.0
	2	220	220	100.0	0.0	0.0
	3	245	245	100.0	0.0	0.0
	4	354	354	100.0	0.0	0.0
	5	287	287	100.0	0.0	0.0
	6	495	376	24.0	0.0	76.0
	7	487	363	25.5	0.0	74.5
	8	462	359	0.0	22.3	77.7
	9	452	359	20.6	0.0	79.4
	10	449	372	17.1	0.0	82.9
	11	386	287	25.6	0.0	74.4
	12	353	299	100.0	0.0	0.0
	13	379	278	26.6	0.0	73.4
20 June 2019	1	867	229	47.2	26.4	26.4
	2	434	274	100.0	0.0	0.0
	3	610	324	100.0	0.0	0.0
	4	494	304	100.0	0.0	0.0
	5	269	269	0.0	100.0	0.0
	6	183	183	100.0	0.0	0.0
	7	180	180	100.0	0.0	0.0
	8	193	193	100.0	0.0	0.0
	9	366	342	100.0	0.0	0.0
	10	345	345	100.0	0.0	0.0
	11	379	379	100.0	0.0	0.0
14 August 2019	1	1040	236	68.1	0.0	31.9
	2	979	238	100.0	0.0	0.0
	3	984	235	76.1	0.0	23.9
	4	1168	468	19.9	40.1	40.0
	5	1087	456	16.1	42.0	41.9
	6	1230	499	40.6	0.0	59.4
	7	450	350	22.2	0.0	77.8
	8	437	339	0.0	22.4	77.6
	9	442	339	23.3	0.0	76.7
	10	307	287	100.0	0.0	0.0
	11	311	285	100.0	0.0	0.0
	12	307	296	100.0	0.0	0.0

Note: Significance level α = 0.05; The shadowed image IDs indicate the repeated imaging of certain scenes.

**Table 4 sensors-20-04980-t004:** Root mean square errors (RMSEs) of different crack width ranges.

Crack Width Range (mm)	RMSE Obtained on (mm)	
	12 March 2019	30 April 2019
6~10	±1.8	±2.2
11~15	±2.3	±2.2
15↑	±1.7	±2.2

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
