# Peer review of "Assessment of Cracking Widths in a Concrete Wall Based on TIR Radiances of Cracking"

_sensors, 2020, doi:10.3390/s20174980_

Round 1

Reviewer 1 Report

Majors:

1.
"multi-temporal" should be explained or changed (multiple measurements ?). In my opinion it is only temporal, because time domain is one-dimensional.

2.
Series of images should be shown with additional details for images for the depiction of the crack evolution process.

3.
"Wavelet-based image denoising is useful to smooth the TIR signals and facilitate the calculation of environmental temperature gradients."
and
"Different kinds of filters based on the spatial or frequency domains of images have been proposed by many works for the denoising of visible, infrared, computed tomography, and magnetic resonance imaging"
but in this paper is lack of assessments of such filters. Why "three-level 2-D Daubechies wavelet" is good, because there is no answer in this paper.

4.
There are too many details, like "values in image ID 9 on March 12". Such process should be explained using good plots, not by the using of long text blocks.

"This result illustrated that at a lower temperature or RH the cracking caused an obvious temperature distribution difference on the concrete wall." Simply: add a mark on figure for better support of statements and add reference in text.
There are many such cases.

Minors:

Eq.1 Please fix size of brackets
Linear model sometimes uses x,y and sometimes X,Y in this paper (l.473, Fig.8)
r^2 and r_2 are to fix (l.476, abstract) for example

Reviewer 2 Report

This paper present a method to asses cracks in ancient building using thermal infrared radiatinces. Overll the paper is well written and easy to understand. I just have a minor comment on section 4.3. I would recommend to add noisy and denoised image for clear understanding and also some more graphical explanations. Also please correct some grammatical mistakes e.g line 200, 175, etc.

Reviewer 3 Report

IN SUMMARY:

This referee has found many critical issues that lead to some confusion in the presentation of the research. Moreover, the manuscript is affected by many punctual errors and wrong concepts regarding the background physics of thermography, the principle of the thermography technique, and the background of sensor technology, as shown in detail later (for author convenience) in the referee’s comments.

Moreover, there are critical issues in the sense and the aim of the proposed method itself, as detailed in comments below.

For these reasons, the opinion of this referee is that the manuscript, as such, can’t be published in a journal strictly related to SENSORS.

Authors of the paper “Assessment of Cracking Widths in A Concrete Wall Based on Environmental Thermal Infrared Radiances” are not presenting any novel thermal method for the detection of cracks, as claimed in the abstract/conclusion. Instead, they are processing vis-thermal data in order to compute in automatic way the crack's width.

Probably they are analyzing data in scrupolous way, from the point of view of processing, but they are misleading the meaning of the thermal data.

I suggest to clarify that it is a computer vision research, and not a non-destructive-technique research.

Perhaps, there are this two possibilities:

- to re-write the paper, by cutting away the parts related to physics and thermography, as these parts are not correct, and by focussing the manuscript on the computer science part that is the key of the work.

- to adjust the wrong parts, by collaborating with some researcher with skills in physics or thermography, as it is evident that the backgrond concetps are missing.

In any case, a manuscript with errors (detailed later in the referee’s comments) cannot be published, as such.

CRITICAL GENERAL ISSUES

There is a simple and critical issue underpinning the method proposed by the authors, for which it is not clear the sense and aim of the research.

They use the thermal images to compute cracks’ width. They extract the crack from the visible, and they input this crack to thermal, in practice. Only surface cracks are studied, for this reason. The width of surface cracks can be computed directly in visible photographs, it is not necessary to pass per thermography. Therefore which is the aim of the method?

What is could be of interest, is the use of thermography to detect the sub-surface cracks (not revealed in the visible range). From this point of view, the research seems just the initial step of the work. Anyway, the more challenging steps, are the calculation of geometrical features, in sub-surface defects, directly from thermograms, as thermograms collect the radiation from the surface of the object, which is the results from the heating of the volume.

Authors themselves in fact specify (line 102) that *sub-surface* cracks is not detected in the visible range and that thermography is not able to measure the morphology; therefore, they applied the procedure "preliminary" to the morphology of *surface* defects.

A general critical issue, here, is that sub-surface thermal analysis is a distinct problem that is very different from surface thermal analysis.

In fact, thermal camera acquires the radiation emitted+reflected from the object (plus, eventually the environment contribution). This radiation can be related to the surface temperature through the Planck law, after that the emissivity of surface materials is given.

When imaging sub-surface defects with a termocamera, the features in the 2D thermogram are the results of the reflected and trasmitted thermal waves inside the material stratigraphy: the analysis of the temperature of the inner defects from the surface temperature, and/or the analysis of the inner morphology from the surface temperature are very difficult inverse problems.

Authors seem to ignore the basic principle of thermography.

CRITICAL SPECIFIC ISSUES

A specific critical problem regarding the method is that the TIR images acquired by the NEC thermocamera are not radiometric.

A thermocamera records the TIR radiation emitted+reflected by the object, (the eventual contribution of the intermediate environment here is negligible, as working distance is short). The emitted radiation is the blackbody radiation by an object at a given temperature; the reflected radiation is given by the environments and kirkhoff laws.

The emitted radiation, without the spurious reflection, is converted to temperature through a calibration of the inband (in the LWIR range) blackbody contribution. This calibration can be made by in-house pre-processing by the camera, and temperature values can be eventually given as grey level intensity values (that are computed, and not recorded by the sensor), but this conversion depends on the emissivity of the materials, which must be given as input in the camera, and on the reflection contribute, which must be also given as input in the camera, usually as "reflected temperature".

Autors do not mention anything about this, and treat the thermal image as simple recorded temperature.

This is not correct in sensor physics.

In their case study, for example, the emissivity of the concrete is different than emissivity of the air inside the crack. Moreover, emissivity depends also on the surface of the materials; if the concrete is deteriorated by chloride penetration, emissivity is different.

In order to extract true temperature values from a TIR image, the emissivity of the object must be firstly measured. There are known method to do this, by using reference emissivity material.

The emissivity problem is a known critical issue in thermography. If active thermography is used, features can be extracted from the cooling/heating curve, without emissivity input. But when a single image is used to extract the measure of the temperature, i.e. the thermocamera is used to measure the temperature, the emissivity problem must be considered.

Otherwise, the values of the temperature are not correct.

Authors compute the temperature gradient, in a non-homogenous medium (different emissivity), therefore they are calculating wrong T values.

As they used they temperature to calculate width of the crack, the entire method is not correct in the principle.

As the whole research is based on temperature extracted by the TIR image, I suggest to conduct some laboratory research on sample test to study this aspect.

SOME PUNCTUAL ISSUES FOR YOUR CONVENIENCE

ERRORS / WRONG USE OF TERMINOLOGY:

1) In the NDT and thermography community, "TIR" is the accepted acronym for Thermal InfraRed, and not for Thermal Infrared Radiance.

In the paper, it is used sometime in a way, sometime in the other.

I suggest to avoid misleading, and to follow widespread used terminology.

2) In science and thermography community, "environmental TIR radiances" means the contribution of the environment, i.e. the thermal blackbody energy radiated by the environment between the camera and the object that contributes to the whole signal collected by the thermocamera.

Adjust this error.

3) Similarly, the term "environmental temperature" is used by the authors to indicate the temperature of the object (crack). Instead, to be correct, this term means the temperature of the environment.

Of course, the object is stressed by the environment, it is not in static equilibrium, it absorbs energy from solar irradiation, its temperature is different from that of the environment.

Adjust this error.

4) For the same reason, the acronym ETD = environmental temperature difference, used by the authors is not correctly used, as in the paper it indicates differences in the temperature field on the object and not of the environment.

Adjust this error.

INLINE COMMENTS  

ABSTRACT-CONCLUSIONS

Mentioning ultrasound is not pertinent: paper does not regard US, the presented method is not compared to US, and US is not e reference technique but only one of the NDTs used…

Authors claim that a "thermal camera is employed to detect cracking".

Instead, the research uses a visible-thermal camera that captures both the visible photograph and the thermogram and, firstly, detect the surface cracks in the visible photograph, secondly, read the temperature of the crack in the superimposed thermogram.

The crack is not detected by thermography, but by visible photographs.

In the conclusion they claim to apply the method in sub-surface cracks. The method, as such, can't be applied to sub-surface crack, obviously, as they are not detected in the visible photographs. This point is detailed later as referee’s comment to the text.

SECTION 1. INTRODUCTION

1) Lines 89-92 and Figure 2

PAPER:

"If an environmental temperature difference(ETD) between the inside and outline of a concrete crack is significant, the concrete crack in 3D plot will be displayed as a canyon-like shape. A wider concrete crack should be indicated by a wider

canyon-like shape, but it is certain that temperature gradient must exist along with ETD"

COMMENTS (MINOR):

surface / mesh plot is only the graphic visualization of the scalar temperature field T(xy). The 3D aspect here has clearly no physical meaning.

As the crack represented in the 2D thermal imaged are 3D features in the real world, I suggest to delete this phrase because it creates misleading. Moreover it is not useful.

2) Line 95

PAPER:

"..regarding non-destructive testing applications, a TIR camera, compared to ultrasound-excited thermography or other heating-wise detection systems, is more accessible and portable."

COMMENTS (PRECISATION):

Adjust, as it is not correct.

In every active thermography (e.g. US-excited) there is a TIR (=Thermal InfraRed) camera. What is different is only the use of an external (artificial) stimulus. In the thermography and science community in general the correct names are "active" or "passive" thermography.

Both acquire the thermal radiance from the object using the thermal camera, the first after an artificial stimulus, the second after the environment (solar, daily cycles) stimulus

3) Lines 104-108

Regarding the three "important contributions" of the paper claimed by the authors, these referee has some comments.

PAPER:

" to investigate the performance of crack detection according to the image processing and statistical analysis results of the acquired TIR dataset;"

COMMENTS (CRITICAL):

It is not true that cracks are detected in thermal images.

Authors use the visible images to identify and segment the cracks, and then use the thermal images to read the temperatures on the related pixels.

They use a camera that acquire simultaneously a pair of superimposed visible and thermal images. They detect cracks in visible photographs, this is the reason why they can apply the method only to surface (i.e. visible) cracks and not to sub-surface (not visible) cracks.

I suggest to clarify across the paper that a VIS-TIR dataset is used (the VIS to detect, the TIR to compute some properties of the cracks) to avoid ambiguity.

This could also help in promoting the results of the work, which concern the data processing and not the NDT method.

On this aspect, the title of the paper "Assessment of Cracking Widths in A Concrete Wall Based on Environmental Thermal Infrared Radiances" is also not clear.

PAPER:

" to determine the relationship between the crack widths and the environmental temperature gradients (ETGs) of the crack sections; "

COMMENTS (CRITICAL):

the object is a non-homogeneous structure, i.e. a concrete wall with cracks, where the discontinuities (cracks) are thermal resistive defects (voids) on the surface.

Under the effects of the environment (daily cycles, plus seasonal cycles) the object is subject to heating and cooling phases, and its temperature is the results of the thermodynamic equilibrium. The equilibrium is a complex dynamic process that depends from lot of factors: beside the environment, the absorbance of the material.

Authors seem to ignore the complex factors regarding thermography and thermal data acquired on real object on field.

PAPER:

" and to validate the precision of applying the environmental TIR dataset to the crack width estimation. "

COMMENTS (CRITICAL):

Authors use thermocamera as simple temperature sensor, but it is not correct. This point is detailed later as referee’s comment to the punctual text.

SECTION 2.1 - STUDY SITE

4) Lines 127-131 and Figure 3

PAPER:

"According to the meteorological data ... this paper analyzed the frequency (day) of the temperature differential, as shown in Figure 3 ...."

COMMENTS (CRITICAL):

Authors plots whether data by the CODiS, Taiwanese Central Weather Bureau the and discuss them in relation to the investigated object.

As explained above, the object response is in relation with the local micro-climate parameters. First important issues is the exposure to direct solar irradiation for example, as the absorbed energy strongly depend on the incident flux (surface-source orientation).

I suggest to avoid simplicistic discussion of the environment data, in detail. Otherwise, a temporal serie of measurement (on the object) should be taken.

SECTION 2.2 - TIR DATASET

5) Line 134-136

PAPER:

"Due to the different lenses needed for TIR and visible RGB images, the spatial resolutions and the principal points of the two kinds of images are also different."

COMMENTS (CRITICAL):

This is completely wrong in physics.

The different spatial resolution is not due to the lens, but to the wavelength of the radiation (different diffraction limit), and mainly, here, it is due to the small size of the thermal sensor.

Authors seem to ignore how an imaging system works.

6) Line 139

PAPER:

Thermal sensitivity ... 0.05 °C.

COMMENTS (PRECISATION)

The AVIO NEC datasheet reports a lower nominal sensitivity of 0.2/0.1 degree.

Did authors measure this different sensitivity in laboratory?

7) Line 146-147

PAPER:

"The specifications of the TIR camera indicated that the FOV would expand as the focusing range (i.e., the object distance) increased [36]."

COMMENTS (PRECISATION)

This is obvious in any imaging system.

I suggest to avoid enriching the paper with non informative and obvious sentences because they are distracting. This paper needs to be focused on the work done, instead.

8) Line 156-162

PAPER:

"Temperature and relative humidity are the main factors related to the process of weathering. As the season changes from spring to summer, the temperature and RH gradually rises. The temperatures on March 12 and April 30 were higher than that on May 10; however, this may have been due to the TIR dataset acquired in afternoons. The interaction of temperature and RH during the different seasons promotes the physical weathering of masonry materials [38-40], thus influencing the inside as well as the outside temperatures of a concrete crack."

COMMENTS (CRITICAL):

As said, equilibrium T and RH are related to local conditions on the object and it is not possible to discuss influence of the city/regional weather in general.

Authors write that the TIR images acquired in different hours of the day are difficult to compare.

During the phase of acquisition, they should have taken into account this fact, by acquiring different series of data during the day. An analysis of variation of thermal radiance during the day can be performed, while an analysis of punctual termogram acquired monthly is not reliable.

SECTION 3 - METHODOLOGY

9) Line 173

PAPER:

"The image segmentation applied a 5×5 weighted median filter..."

COMMENTS (PRECISATION):

Suppose that unit is pixels (should be always given). Therefore, as they use different working distance, this means different size of pixel at object plane, and different level of noise.... while using the same windowing?

10) Lines 179

COMMENTS (CRITICAL):

As explained above, it is not possible to simply extract temperature values by a thermal image without inputting the emissivity values of the object.

SECTION 3.1 – IMG REGISTRATION

11) Line 188

COMMENTS (MINOR)

The authors are using a commercial SW to geometrically calibrate the VIS-TIR cameras.

I suggest to not enter in such details of the theory behind.

Instead: geometrical calibration is performed on the hypothesis of a pinhole model vision system. I suggest to compute also the distortion.

12) Line 244

COMMENTS (PRECISATION):

Please specified the resolution (pixel array) also of the RGB sensor.

SECTION 3.3 – STATS ANALYSIS

13) COMMENTS (CRITICAL):

Descriptive statistics is used to analyze thermal data.

Anyway, they are not performing the analysis on temperature data. As it seems that they have not calculated the temperature from radiometric data using the emissivity matrix.

SECTION 3.3 - DENOISING

14) Line 305-312 and formula

PAPER:

..pixel values in TIR image is the result of a thermal infrared intensity measurement made by a CCD matrix... ????

...In each captor of the CCD matrix, the number of incoming TIRs is counted .... ????

... number of TIRs ?????

...Obscurity noise...???

...observed temperature value v(i) ?????

... recorded temperature values ????

... spurious TIRs from sensed object ????

COMMENTS (CRITICAL):

This part is completely wrong in physics of sensors.

Authors have taken the part written in reference [46], regarding CCD sensors, then they have simply replaced what was referred to VISIBLE LIGHT PHOTONS with "TIRs".

This can't be done for the following reasons:

- CCD are sensitive up to 1 micron. Thermal camera does not mount a CCD sensor, as they write. Thermal camera use different principle and different technology (mainly bolometers) to acquire the thermal photons in the LWIR range.

This error can't published, especially in a journal focused on SENSORS.....

- "incoming TIRs", "number of TIRs", means nothing. Camera records Thermal InfraRed (TIR) PHOTONS.

- the correct term is Dark Noise and not "obscurity"

- again, the camera does not record temperature values.

Thermography record total infrared radiation (PHOTONS), which must be related to temperature value by calculation.

This calculation is not linear (plank law), the noise is not simply an additive noise to the temperature value, like the dark noise of the visible camera.

- It is not true to say that the thermocamera record spurious TIRs (photons) by the sensed object. A spurious fluctuation is given by the electronics, as written also in [46] for RGB camera. The heating of the sensor due to electronics of course is a more critical problem for thermal camera.

MINOR COMMENTS, FOR YOUR CONVENIENCE

Line 35

Wrong use of term “resistivity”. Probably you mean resistance, that is a different concept.

Line 121

"Self dead load" ???

adjust: Dead loads or self weight, they are the same.

Line 127

"Multi-temporal" ???

is not clear, use just temporal.

Line 190

Suggesto to put the reference of the commercial SW used for image registration, as in internet it is not possible to find any information on this regard.

COMMENT TO BIBLIOGRAPHY

An estensive bibliography is given regarding lot of topics, which result distracting. For example, literature on US technique is given. Anyway, US is not concerned with the paper, and on the contrary, specific bibliography on infrared thermography applied to the study of cracks is poor.

I suggest to adjust the focus of the cited references discussing only or majorly the NDT literature related to the method used in the paper.

Reviewer 4 Report

This paper presents interesting results but needs a thorough revision before being considered for publication. Some sections need to be completely rewritten, like the Introduction, literature review and Discussion.
Introduction: The theoretical, analytical and standard approaches should be discussed.
The novelties have to be outlined. It has to be completely rewritten so that the focus of the work and its innovative content can be really appreciated.
Literature review: The Literature review is now a mere list of information but the authors have to provide their own "unifrying" view and not only citing previous work.
In addition, in a quick search I found a number of papers on this topic that you did not cite. 

Results and discussion: The paper presents a big amount of results from usual experiments but without a theoretical and practical approach.
Conclusions: The discussion about technological benefit have to be separated in the article according points of conclusions. The analysis of the results is quite basic and deserves better and deeper processing.

Round 2

Reviewer 1 Report

ok

Author Response

Thanks for your review.

Reviewer 3 Report

The authors have answered almost every referee's request of clarification and have corrected the errors.

This referee is still convinced that the underpinning methods lacks of some basis and suggest to the authors to consider this work as the first stage of future research.

At this stage, this referee leaves to the editor and authors the freedom of improving the remaining issues, as everything have been said.

(TIR errors)

- Some errors regarding the use of TIR are still present. 

LINE 529 and LINE 545 (spectral resolution ???).

(abstract)

For your convenience:

suggest you to think about the fact the the effective of working distance depend on the size of the sensor array, therefore the minimun focal length should be refereed to that sensor array, it is not general.

(feedback on comments)

For your convenience:

the comment 18 was not regarding the pixel unit,  but it was a suggestion to think about the true size of the pixel at object plane and how to correct the noise level with an optimized window size. Anyway, I suggest to consider that it is a good practice to specify the units, especially in engineering works related with real world applications.

(general method)

For your convenience:

automatic correction emissivity is a non sense, as the emissivity depends on specific materials.

Authors now answered that they are computing only difference in temperatures. Anyway, if you convert radiation to temperature on different materials, as air and concrete, the emissivity parameters cannot be factorized and the difference in temperature definitively depends on the emissivity.
